# A Novel Attention-Mechanism Based Cox Survival Model by Exploiting Pan-Cancer Empirical Genomic Information

**DOI:** 10.3390/cells11091421

**Published:** 2022-04-22

**Authors:** Xiangyu Meng, Xun Wang, Xudong Zhang, Chaogang Zhang, Zhiyuan Zhang, Kuijie Zhang, Shudong Wang

**Affiliations:** 1College of Computer Science and Technology, Qingdao Institute of Software, China University of Petroleum, Qingdao 266580, China; xiangyumeng@s.upc.edu.cn (X.M.); wangsyun@upc.edu.cn (X.W.); bigdongsir@163.com (X.Z.); s20070030@s.upc.edu.cn (C.Z.); flyeagle237@163.com (Z.Z.); z20070009@gmail.com (K.Z.); 2China High Performance Computer Research Center, Institute of Computer Technology, Chinese Academy of Sciences, Beijing 100190, China

**Keywords:** deep learning, survival analysis, neural networks, Cox regression, cancer prognosis

## Abstract

Cancer prognosis is an essential goal for early diagnosis, biomarker selection, and medical therapy. In the past decade, deep learning has successfully solved a variety of biomedical problems. However, due to the high dimensional limitation of human cancer transcriptome data and the small number of training samples, there is still no mature deep learning-based survival analysis model that can completely solve problems in the training process like overfitting and accurate prognosis. Given these problems, we introduced a novel framework called SAVAE-Cox for survival analysis of high-dimensional transcriptome data. This model adopts a novel attention mechanism and takes full advantage of the adversarial transfer learning strategy. We trained the model on 16 types of TCGA cancer RNA-seq data sets. Experiments show that our module outperformed state-of-the-art survival analysis models such as the Cox proportional hazard model (Cox-ph), Cox-lasso, Cox-ridge, Cox-nnet, and VAECox on the concordance index. In addition, we carry out some feature analysis experiments. Based on the experimental results, we concluded that our model is helpful for revealing cancer-related genes and biological functions.

## 1. Introduction

Since the 20th century, cancer has become a serious threat to human life and health. Multi-angle and multi-level prognostic studies on cancer emerge one after another. However, prognostic research for cancer is still a challenging task. One of the most important factors is that due to the existence of censored patient samples, the traditional analysis models cannot effectively determine the actual time of death [1]. Compared with traditional prognosis, survival analysis models focus more on the survival time point of the patient rather than the death time point. The use of survival analysis models can be very effective in dealing with censored data. The most widely used is the Cox proportional hazards model (Cox-ph) model [2]. The Cox-ph model is a semi-parametric proportional hazards model. The covariates of the model can explain the relative risk of a patient, called hazard ratio, prognostic index or risk score. According to the relative risk score, the risk of changes in various factors can be effectively analyzed, thereby helping doctors to develop effective targeted therapy strategies.

The arrival of the Human Genome Project has raised the survival sub-model to the research category of high-throughput multi-omics [3]. There are many experiments and studies showing that it is very meaningful to do cancer survival analysis on high-throughput transcriptome data [4,5,6]. Computationally, however, performing survival analysis on high-dimensional transcriptome gene expression data is equivalent to solving a regression problem of complex nonlinear equations. Using traditional Cox-ph regression models cannot effectively handle high-dimensional feature representation and accurately predict prognosis. In the past two decades, some researchers have used some machine learning methods to modify the original Cox survival analysis model, and the regression effect of the model has been improved to a certain extent. Some methods use the support vector machine (SVM) algorithm to perform feature extraction and dimensionality reduction for high-dimensional gene expression data [7,8]. The result after mechanized dimensionality reduction of SVM fuses the original high-dimensional gene features, and the use of Cox-ph model to represent low-dimensional gene expression features can effectively implement survival analysis prediction. Some methods will use an ensemble learning method such as Cox-Boost, which divides the parameters into several independent partitions for ensemble training and fitting [9]. Some methods replace the original proportional hazards model by using the ensemble mean cumulative hazard function (CHF) with the help of the nonlinear ensemble method of random forests [10].

With the maturity of deep learning methods in different fields [11,12], the Cox model, based on an artificial neural network, has received extensive attention from researchers. To the best of our knowledge, the earliest application of artificial neural networks for survival analysis is Faraggi et al. [13]. They used four diagnoses as inputs to model the learning of a survival analysis for prostate cancer. Then Ching et al. designed a Cox-nnet composed of two-layer neural networks, and successfully used Cox-nnet to make reasonable survival analysis recommendations for 10 different cancer gene expression data [14]. Katzman et al. proposed a Cox regression model constructed by a multi-layer neural network, while formulating corresponding treatment recommendations based on the trained Cox model [15]. Huang et al. [16] proposed a survival analysis model for multi-omics data of breast cancer. They first constructed a co-expression feature matrix of mRNA data and miRNA data through gene co-expression analysis to alleviate the learning difficulties and overfitting problem caused by high-dimensional data. They finally proposed a multi-group student survival analysis model for breast cancer [16]. Kim et al. used a transfer learning approach, first pre-training a VAE model, and then using the VAE model for fine-tuning training on 20 TCGA datasets [17]. Using the above method effectively alleviates the overfitting problem caused by the high genetic dimension and the small number of training patients. Ramirez et al. [18] focused on the degree of correlation between different genes. They constructed gene association graphs through correlation coefficient scores, PPI networks and other methods, introduced the correlation map as a prior knowledge into the training of the GCN survival analysis model, and explained the important biological significance of the GCNN model in survival analysis [18].

The high dimensionality and complex semantics of genetic data bring many challenges to feature extraction. So far, there is a lot of work that is trying to use new ideas to ensure rich feature extraction. The most classic method in deep learning was multilayer perceptron (MLP), which learns linear correlations between different data. It was considered to be the optimal choice for processing sequential data. There are a lot of works to extract high-dimensional genetic data based on MLP [14,16,19]. In the past decades, convolutional neural networks (CNN) have shown excellent results in computer vision. More and more studies have demonstrated the powerful ability of CNNs to deal with spatial structural features. Many recent works have attempted to extract features from high-dimensional genetic data using CNNs and demonstrated the transferability of CNNs in multi-omics data. Rehman et al. proposed densely connected neural network based N4-methylcytosine site prediction (DCNN-4mC), and this framework obtains the greatest performance for 4mC site identification in all species [20]. Chen et al. used Lasso and CNN as a target model and studied the trade-off between the defense power against MIA and the prediction accuracy of the target model under various privacy settings of DP [21]. Torada et al. proposed a CNN-based program, called ImaGene, on genomic data for the detection and quantification of natural selection [22]. Hao et al. proposed a biologically interpretable deep learning model (PAGE-Net) that integrates histopathological images and genomic data [23]. Jeong et al. proposed a new tool called GMStool-based CNN for selecting optimal marker sets and predicting quantitative phenotypes [24]. Rehman et al. proposed the m6A-NeuralTool to extract the important features from the one-hot encoded input sequence based on CNN [25]. At the same time, some works [18,26] take advantage of novel feature extraction methods for sequence data and exhibit remarkable results.

Since the Generative Adversarial Network (GAN) proposed by Ian et al. [27], a lot of research has used this training strategy for data generation, reconstruction and dimensionality reduction tasks. GAN adopts an adversarial training strategy, which effectively learns the distribution of high-dimensional data and effectively generates results sampled from the overall data distribution. In recent years, GAN has been widely used in protein and gene sequence research. Repecka et al. [28] developed a variant of attention-based GAN and called it ProteinGAN. ProteinGAN learns the evolutionary relationships of protein sequences directly from the complex multidimensional amino acid sequence space and creates highly diverse new sequence variants with natural physical properties, which demonstrates the potential of GANs to rapidly generate highly diverse functional proteins within the biological constraints allowed by the sequence space. LIN et al. [29] proposed the DR-A framework, which implements dimensionality reduction for scRNA-seq data based on an adversarial variational autoencoder approach. Compared with traditional methods, this method can obtain a low-dimensional representation of scRNA-seq more accurately. Jiang et al. [30] introduced a novel GAN framework for predicting disease genes from RNA-seq data. Compared to state-of-the-art methods, the model improves the identification accuracy of disease genes.

In this paper, we propose a novel deep Self Attention Variational Autoencoder Cox Survival Analysis Model (SAVAE-Cox). This model takes advantages of adversarial transfer learning strategy. In the adversarial pretraining stage, the generator was a variational autoencoder (VAE), which is jointly trained with the discriminator. Meanwhile, we introduce a novel self-attention mechanism [31] to enhance semantically relevant features extraction of the encoder from high-dimensional data. After the pretraining stage, the generator was able to learn the common features of 33 cancer transcriptome data. Next, the encoder of the generator was used to learn survival analysis on 16 cancers. By comparison with state-of-the-art models such as Cox-nnet and VAECox, our model achieved the highest concordance index on 10 TCGA cancer datasets. Finally, we performed feature analysis of SAVAE-Cox. We select oncogenes and compute correlations with hidden layer nodes in which we find that our hidden layer nodes are highly correlated with oncogenes. We used these nodes to draw Kaplan–Meier plots, and found that these nodes significantly affected the survival of patients. Based on the correlation of hidden layer nodes with genes, we selected leader genes, which we found enriched on cancer-related pathways. According to our experiments, we conclude that our proposed SAVAE-Cox model has significant cancer prognostic ability. Our source code of SAVAE-Cox is available at https://github.com/menggerSherry/SAVAE-Cox (Last visited on 21 March 2022).

## 2. Materials and Methods

### 2.1. Dataset Preparation

In this work we used 17 datasets from the TCGA database. These 17 datasets are bladder carcinoma (BLCA), breast carcinoma (BRCA), head and neck squamous cell carcinoma (HNSC), kidney renal cell carcinoma (KIRC), brain lower-grade glioma (LGG), liver hepatocellular carcinoma (LIHC), lung adenocarcinoma (LUAD), lung squamous cell carcinoma (LUSC), ovarian carcinoma (OV), stomach adenocarcinoma (STAD), cervical squamous cell carcinoma and endocervical adenocarcinoma (CESC), colon adenocarcinoma (COAD), sarcoma (SARC), uterine corpus endometrial carcinoma (UCEC), prostate adenocarcinoma (PRAD), skin cutaneous melanoma (SKCM) and pan-cancer (PANCAN). The detailed description of the dataset and download way were in Section A.1. Since there are a large number of empty genes and noise genes, it is necessary to perform some preprocessing options to exclude some redundant noise genes. In Table 1, we present the statistics of the 17 datasets used in this work, including the number of samples in each dataset and the clinical information of 16 cancer types.

We use the PANCAN dataset to draw scatter plots of 56,716 genes and observe the statistical distribution. Figure 1 shows the distribution of mean and standard deviation of RNA-seq data. From the standard deviation distribution of Figure 1, we observe that there is a valley between 0.278–0.403, and plenty of genes are with zero variance. Therefore, we defined gene expression with a standard deviation in the range (0, 0.4) as noise genes, and removed those noise genes whose standard deviation satisfies this range. At the same time, in order to eliminate the influence of empty data, we removed the RNA-seq genes whose mean value satisfies (0, 0.8). We processed each RNA-seq dataset following these two strategies described above and selected 20,034 intersection genes of 16 cancer types. Before feeding the genes into our module, we performed feature wise min-max normalization of each gene of 16 cancer types.

### 2.2. Dimensionality Reduction Pretraining Using GAN

We adopted the strategy of a generative adversarial network (GAN) [27] to design the pre-training stage. The pre-training process is shown in Figure 2a. The generator G takes the genes $x_{in}$ as input and generates the reconstructed genes $x_{rec}$. The discriminator D takes the $x_{in}$ or $x_{rec}$ as the input and outputs a value which reflects the authenticity of the gene. Through adversarial training with the D, the encoding module E of G gradually improves the feature extraction ability to generate a low-dimensional feature z for $x_{rec}$ generation. After training, E can be used for dimensionality reduction of $x_{in}$. Compared with computational dimensionality reduction methods, our dimensionality reduction based E was a data-driven method and can be adaptively adjusted according to different data characteristics.

The network structure of the G is a self-attention VAE(SAVAE) framework. For each $x_{in}$, SVAE is described as:(1)$$\mu \left(x_{in}\right)=w_{\mu }\left({ℒ}^{\alpha }\left(\delta \left(w_{h}x_{in}+b_{h}\right)\right)\right)+b_{\mu }$$
(2)$$\sigma {\left(x_{in}\right)}^{2}=exp\left(w_{\nu }\left({ℒ}^{\alpha }\left(\delta \left(w_{h}x_{in}+b_{h}\right)\right)\right)+b_{\nu }\right)$$
(3)z=μζ+σ,ζ∼N(0,1)
(4)$$x_{rec}=w_{h}z+b_{h},$$
where $\mu \left(x_{in}\right)$ and $\sigma {\left(x_{in}\right)}^{2}$ are the mean and the variance of the Gaussian disribution. δ is the activate function. ζ is randomly sampled from the standard Gaussian distribution.

We introduce a residual self-attention module (Figure 3) in the hidden layer of the encoder to enhance the fitting ability of VAE [32]. The self-attention mechanism [31] can effectively learn the semantic correlation of high-dimensional features. It is denoted as:(5)$${ℒ}^{\alpha }\left(x\right)=sa\left(x\right)+\alpha \times x.$$

In Equation (5), α represents a learnable parameter, which can adaptively adjust the weight of the residual connection. sa stands for Self-Attention Module [31]. It is denoted as:(6)$$sa\left(x\right)=softmax(Q\left(x{)}^{T}\times K\left(x\right)\right)\times V\left(x\right),$$
where Q, K, and V represent the query, key and value obtained by performing three Dense layers on the input x.

D is a simple binary classification network whose framework is represented as follows:(7)D(x)=ℒoutd⊙ℒnd⊙ℒn−1d⊙⋯⊙ℒ1d(x),
where the ℒnd:ℝN→ℝN2 maps high-dimensional features to low-dimensional features through linear transformation. Finally, the output feature of ℒnd was fed to a classification layer ℒoutd:ℝN2n→ℝ. The classification layer generates a numerical value, which represents the judgment of the input gene.

We introduce the Wasserstein loss [33] to train G and discriminator D jointly. For G, the goal is to synthesize more similar reconstructions $x_{rec}$ so that the discriminator cannot describe whether it is real or fake. For D, the goal is to distinguish between true genes $x_{in}$ and reconstructed genes $x_{rec}$ synthesized by G. The loss of G and D is computed as:(8)$${ℒ}_{gan}=E_{x_{in}∼P_{data}\left(x_{in}\right)}\left[D\left(x_{in}\right)\right]-E_{x_{in}∼P_{data}\left(x_{in}\right)}\left[D\left(G\left(x_{in}\right)\right)\right]-\lambda_{p}E_{x∼X}{\left[|\left|\nabla_{x}D\left(x\right)\right|{|}_{2}-1\right]}^{2},$$
where $\lambda_{p}$ represents the hyper parameter for setting the gradient penalty, and X represents the overall Sample Space of $x_{in}$ and $x_{rec}$. The Wasserstein loss was calculated to minimize the Wasserstein distance between $x_{in}$ and $G\left(x_{in}\right)$ by the D, which made the overall sample distribution of $x_{in}$ and $G\left(x_{in}\right)$ more similar. Furthermore, we introduce the Kullback–Leibler divergence [32] used in training the VAE. Given the input Genes $x_{in}$. it is denoted as:(9)$${ℒ}_{KL}=\overset{n}{\underset{i=0}{\sum }}(\mu {\left(x_{in}\right)}^{2}+\sigma {\left(x_{in}\right)}^{2}-log\left(\sigma \left(x_{in}{)}^{2}\right)-1\right),$$
where n represents the dim of z. This error measures the distance between the real latent code z under standard Gaussian distribution and the posterior latent variable $P(z|x_{in})$ generated by encoder. The introduction of Kullback–Leibler divergence into the generator can guarantee the similarity between low-dimensional latent variables, which significantly improves the authenticity of the distribution of $x_{in}$ and $G\left(x_{in}\right)$. At the same time, we introduce *L*1 loss to measure the similarity of each sample. The *L*1 loss is computed as:(10)$${ℒ}_{L1}=|\left|x_{rec}-x_{in}\right|{|}_{1}.$$

Unlike the Wasserstein loss and KL divergence, the *L*1 loss focuses on making G learn to synthesize $x_{rec}$ more similar to $x_{in}$ on genes-wise. Therefore, the overall loss function for G can be expressed as:(11)$${ℒ}_{total}={ℒ}_{gan}+\lambda_{1}{ℒ}_{KL}+\lambda_{2}{ℒ}_{L1},$$
where $\lambda_{1}$ and $\lambda_{2}$ represent the hyper parameters. We therefore aim to solve:(12)$$\theta_{G}^{*}=arg\underset{G}{\min}\underset{D}{\max}{ℒ}_{total}.$$

### 2.3. Survival Analysis Based on Transfer Learning

After the pre-training stage, we transfer the weights learned by the encoder in the pre-training stage to the survival analysis stage as shown in Figure 2b. At the same time, we set an additional classification module to learn the hazard ratio. The hazard ratio measures the likelihood a patient has of dying, and a higher hazard ratio indicates a higher likelihood that a patient will die.

We adopt the training strategy of Cox-ph [2] to train our module, which is denoted as:(13)$$h(t|x_{i})=h_{0}\left(t\right)exp\left(wx_{i}\right),$$
where $h_{0}\left(t\right)$ was baseline hazard function, w was the trainable parameters of the module, and $x_{i}$ represents the risk factors of patients. At this stage, $x_{i}$ is the low-dimensional feature $\mu \left(x_{in}\right)$ that output by SAVE encoder. We aim to solve:(14)$$\theta^{*}=arg\underset{\theta }{\min}\underset{C\left(i\right)=1}{\sum }(wx_{i}-log\underset{t_{j}\geq t_{i}}{\sum }wx_{j}),$$
where t is the the survival time of patient sample, C(i) indicates whether the patient sample i is censored.

### 2.4. Experiment Settings

We implemented our model using the PyTorch framework. We train and validate our model using an Nvidia Tesla V100 (32GB) GPU. We first pre-train SAVAE using the pan-cancer database. To validate the pre-trained reconstruction results, we randomly divide the training dataset and the test dataset with a ratio of 9:1. For survival analysis on 16 cancer datasets, we trained our model by dividing the dataset into five-fold cross-validation. Both stages are trained using the Adam optimizer. In the pre-training stage, the learning rate of the optimizer is 0.0001, the total epochs are 300, and the batch size is 256. At the same time, since the loss fluctuation problem often occurs in the training process of GAN, it is necessary to set the learning rate decay strategy. Therefore, we stipulate that the learning rate remains constant for the first 150 epochs, and decays linearly to 0 for the last 150 epochs. In the SAVAE-Cox training phase, the learning rate of the optimizer is 0.001, the total training epochs are 20, and the batch size is 512. Note that there are some hyperparameters such as learning rate,$\;\lambda_{p}$, $\lambda_{1}$ and $\lambda_{2}$. The specific selection methods and optimal parameter settings are in Section A.2. To ensure the fairness of training, we use the same dataset settings to train and evaluate Cox-nnet, Cox-lasso, Cox-ridge and VAE-Cox.

### 2.5. Evaluation Metric

In this work, the evaluation method we mainly use is the concordance index [34], which is widely used in survival analysis models. It ranges from 0 to 1. When the concordance index ≤ 0.5 means that the model has completed an ineffective survival analysis prediction. When the concordance index > 0.5 and higher this indicates that the prediction effect of the model is better.

## 3. Results

### 3.1. Performance of Dimensionality Reduction

In Section 3.1, we evaluated the generator performance in the pre-training stage. All 990 samples in the pan-cancer test dataset were used in this section. We fed the test set of the pan-cancer dataset to the generator, and the generator synthesized the reconstruction results. We then use UMap to perform visualization of the real genes and the reconstructed genes. Figure 4 is the visualization using UMap. We found that the distribution of reconstructed genes closely coincided with the distribution of real genes, which shows that:
Generator can reconstruct $x_{rec}$ that are consistent with $x_{in}$;The reconstruction of the generator is based on the latent encoding z, indicating that the encoder of the generator can effectively generate z, which retains rich features that can represent $x_{in}$.

To further verify the superiority of our proposed dimensionality reduction method, we compare the performance with other dimensionality reduction methods. The comparison results were shown in Table 2 and the optimal hyperparameter settings were shown in Table A1. First, we choose Autoencoder (AE) and denoising Autoencoder (denoise-AE) to compare with our model. At the same time, we also compare some classical dimensionality reduction methods based on feature selection such as Chi2, Pearson, mutual information and maximal information coefficient (MIC) and principal component analysis (PCA). Note that unlike data-driven dimensionality reduction methods, the feature selection method does not use pan-cancer data for pre-training but directly applies statistical methods on 16 cancer types to select high-correlation features. We used five-fold cross-validation on 16 types of data to train these models and calculated the mean concordance index.

By comparing these six methods on 16 cancer types, our dimensionality reduction method performed best in nine of them. Interestingly, using the Chi2 feature selection-based dimensionality reduction method outperforms data-driven methods in LUSC, CESC, and SKCM datasets. As shown in Table 2, the samples in the CESC and SKCM datasets are small. Meanwhile, we found significant overfitting problems that emerged while training used the data-driven dimensionality reduction methods on the LUSC dataset. These characteristics reflect the shortcomings of data-driven dimensionality reduction methods that are highly dependent on data.

### 3.2. Performance of Survival Analysis

We use the concordance index to evaluate the performance of SAVAE-cox models. Figure 5 shows the comparisons of performance on 16 cancer datasets. In Figure 5, we selected four models to compare with SAVAE-Cox, which include the classic model like Cox-lasso and Cox-ridge methods, as well as some state-of-the-art methods such as Cox-nnet and VAECox. Each model chooses the optimal parameter settings to ensure the fairness of the experiment. Table A1 shows the optimal parameters of these models. Then we divided the 16 cancer types into train and validation datasets by five-fold cross-validation. For each cancer type, we compute the mean concordance index for the validation set. Finally, we draw boxplots according to the concordance index of the five models. From the experimental results we can see that the predicted concordance index using our model on 12 cancer types is significantly higher than the other four models.

We performed further survival analysis on 12 cancer types. Based on the hazard ratios of patient samples predicted by SAVAE-Cox, we can calculate the mean reference hazard ratios for 12 cancer types. For each cancer type, we defined patients with predicted hazard ratios above the average to be in the high-risk group, and patients with predicted hazard ratios below the average to be in the low-risk group. In this way, we divided patient samples into high-risk and low-risk groups for each cancer type. Therefore, we draw Kaplan–Meier (KM) survival curves for different cancer types according to these two groups. At the same time, we adopted the same strategy to plot the KM survival curve of the Cox-nnet prediction results. Figure 6 shows the comparison of KM survival curves of SAVAE-Cox and Cox-nnet on 12 cancer types. Based on the results in Figure 6, we found that the hazard ratio predicted by SAVAE-Cox significantly affected patient survival. At the same time, by analyzing the *p*-value, we can find that the hazard ratios predicted by SAVAE-Cox have a more significant impact on patient survival than that predicted by Cox-nnet.

### 3.3. Feature Analysis of SAVAE-Cox

The incidence of BRCA is very high, so there were a lot of related studies on this malignancy. At the same time, benefiting from the professionalism of the TCGA project, the researchers collected abundant samples. Choosing BRCA for survival analysis can obtain the most stable survival analysis results and conclusions. Therefore, we take BRCA as an example to conduct experiments for further correlation analysis between model features and genes. All of the 1031 patient samples in the BRCA dataset were tested in this study. First, we analyzed the hidden layer nodes that contribute the most to the prognosis according to the mean and variance. After analysis, we selected the top 20 key prognostic hidden layer nodes. Finally, we calculated a Pearson correlations matrix for each node and gene expression across the patient samples. According to the calculated correlation matrix, we can analyze the correlation between hidden layer nodes and different genes.

We selected 34 cancer-related genes from DISEASE (https://diseases.jensenlab.org/, Last visited on 21 March 2022) and plotted them in Figure 7. These genes have high correlation scores in DISEASE, and some genes are oncogenes in ovarian cancer and significantly affect patient survival. The meta-analysis strongly supports the prognostic role of BCL2 as assessed by immunohistochemistry in breast cancer [35]. Germline variation NEK10 is associated with breast cancer incidence [36]. The progesterone receptor (PgR) is one of the most important prognostic and predictive immunohistochemical markers in breast cancer [37]. The CCDC170 gene affects both breast cancer risk and progression [38]. ESR1 amplification may be a common mechanism in proliferative breast disease and a very early genetic alteration in a large subset of breast cancers [39]. The SLC4A7 variant rs4973768 is associated with breast cancer risk [40]. ATM mutations that cause ataxia-telangiectasia are breast cancer susceptibility alleles [41]. RAD51 is a potential biomarker and attractive drug target for metastatic triple negative breast cancer [42]. CTLA-4 was expressed and functional on human breast cancer cells through influencing maturation and function of DCs in vitro [43]. MYC deregulation contributes to breast cancer development and progression and is associated with poor outcomes [44]. The CDH1 mutation frequency affecting exclusively lobular breast cancer [45]. Inherited mutations of BRCA1 are responsible for about 40–45% of hereditary breast cancers [46]. EGFR is an oncogene in breast cancer [47]. ERBB2 is an oncogene in breast cancer [48]. Six different germline mutations in breast cancer families are likely to be due to BRCA2 [49]. Rare mutations in XRCC2 increase the risk of breast cancer [50]. Amino acid substitution variants of XRCC1 and XRCC3 genes may contribute to breast cancer susceptibility [51]. Overexpression of an ectopic H19 gene enhances the tumorigenic properties of breast cancer cells [52]. CYP19A1 genetic variants in relation to breast cancer survival in a large cohort of patients [53]. BARD1 mutations may be regarded as cancer risk alleles [54]. Master regulators of FGFR2 signalling and breast cancer risk [55]. Interestingly, from Figure 7 we can see that 20 key nodes are significantly associated with these oncogenes, which suggests that exploring patient survival using our proposed model could serve as a new avenue for the discovery of oncogenes.

To further explore the contribution of hidden layer nodes to patient survival, we plot Kaplan–Meier survival curves with 20 key nodes. In this experiment, we also selected all patient samples in BRCA for evaluation. By analyzing the variation of each node in the overall patient sample, we divided them into two groups according to the mean value of nodes. Finally, we calculated the log-rank *p*-value for each node and drew the Kaplan–Meier survival curves. Figure 8 shows the survival curves of the first four key nodes. From the survival curve, we found that patient samples can be significantly divided into risk groups and safety groups according to key nodes, and the larger the value of the node, the lower the survival probability of the patient, which proves that our hidden layer nodes can be used as a key prognostic factor.

### 3.4. Biological Function Analysis of Hidden Nodes

To further explore the biological relevance of hidden layer nodes, we performed a gene set enrichment analysis (GSEA) using the KEGG pathway. We ranked genes according to the Pearson correlation and selected the leader gene for each key node. Based on the leader genes, we created the pathway association network (Figure 9). In Figure 9, each point represents a pathway, and the size of the point represents the number of genes enriched in this pathway, which indicates that the hidden layers of our module can effectively learn the biological functions associated with diseases.

### 3.5. Ablation Study for SAVAE-Cox

To verify the contribution of our proposed model to the effect of survival prognosis, we performed ablation experiments on SAVAE-Cox. Briefly, we divided the model into four groups: Cox-nnet, SAVAE-Cox without pretrain, SAVAE-Cox without attention, and SAVAE-Cox. We trained four models on 16 cancer types and divided ranks 1, 2, 3, and 4 in descending order according to the performance of the four models in each cancer and plotted Figure 10. Note that optimal parameter settings were selected for all four models in the ablation study. In Table A1 we listed the parameter settings of these four models.

By comparing SAVAE-Cox and SAVAE-Cox without attention, we find that using the self-attention module can significantly improve the model prognosis accuracy. This result shows that there is a potential feature semantic correlation in high-dimensional gene expression, and this correlation cannot be effectively learned using traditional fully connected layers. This latent relationship can be found to a large extent using attention-based methods. By comparing SAVAE-Cox and SAVAE-Cox without pretrain, we find it interesting that the module that used transfer learning on the five datasets HNSC, KIRC, LUAD, LUSC, and OV fails to improve the model’s prognostic results. In general, using the transfer learning strategy improves the C-index of the model.

## 4. Discussion

In this work, we introduced a novel survival analysis model for different cancer types, which is the first attempt to improve the overall survival analysis accuracy with the help of the self-attention mechanism. At the same time, we designed a data-driven dimensionality reduction method regarding the idea of transfer learning and GAN [27] to further improve the prediction effect. Our results in Figure 5 suggest that the best performance can be achieved using SAVAE-Cox in the prediction of survival analysis for 12 cancer types including BLCA, BRCA, HNSC, KIRC, LGG, LIHC, LUAD, LUSC, SARC, SKCM, STAD, and UCEC. There are multiple factors or complex genetic associations in most of these 12 cancer types that potentially influence patient survival, which shows that the SAVAE-Cox can effectively discover such latent semantic correlations. However, for some small sample datasets like CESC, our model still cannot achieve optimal results. Using transfer learning can indeed alleviate the overfitting problem to a certain extent. However, due to the deepening of the network, it is not comparable to some classical methods in the cancer dataset with sparse samples. Besides, for some cancer datasets like PRAD with an extremely uneven number of positive and negative samples, using SAVAE-Cox for survival analysis is also not the best choice. We analyze that the cause for this problem is also that the complexity of the network increases, which leads to a larger differentiation of the fitting to the two samples, resulting in a loss of accuracy.

We proposed an adversarial VAE-based [32] pre-training method for dimensionality reduction of high-dimensional genes. Unlike classical feature selection methods, our proposed dimensionality reduction method is data-driven. SAVAE-Cox can adaptively extract useful features from high-dimensional genes based on this novel method. This dimensionality reduction method is applicable to any type of data distribution. From Table 2, we found that using the data-driven method showed the best mean concordance index on 13 cancer types, and the effect of using the data-driven dimensionality reduction method was more significant. However, in some aspects, traditional feature selection dimensionality reduction methods may be more effective. For example, in dimensionality reduction tasks for small sample datasets, data-driven methods cannot predict the whole low-dimensional latent space based on a small number of sample distributions. According to the results in Table 2, we can find that using the Chi2 method for dimensionality reduction achieved the best results on two datasets with small sample sized datasets such as LUSC, CESC and SKCM. Meanwhile, we combine GAN [27] and VAE [32] to design a more powerful data-driven dimensionality reduction strategy. This method introduces distribution constraints in GAN, which improves the stability of dimensionality reduction under the condition of ensuring strong generation and fitting capabilities. By comparing the state-of-the-art data-driven dimensionality reduction methods in Table 2, the performance using our method is better.

Through the analysis of BRCA, our model can discover cancer-related genes and reveal biological functions. From Figure 7 and Figure 8 we find that each key node of the hidden layer of the SAVAE-Cox model is a prognostic features affecting patient survival. Furthermore, our model helps to explain and discover new cancer-related genes. Meanwhile, according to Figure 9, numerous node-related genes are enriched in cancer pathways such as the breast cancer pathway, the PI3K-Atk signaling pathway, the Rap1 signaling pathway, and the MPKA signaling pathway, in which we confirmed that the hidden layer of the model is highly related to biological functions and reveal rich biological function signals.

However, we found that the overfitting of the SAVAE-Cox is still a very serious problem. At the same time, the performance of the model is not good enough on datasets with an imbalanced number of positive and negative samples. In future work, we will study some data augmentation methods to solve these problems and explore some novel multi-head attention-based survival analysis frameworks.

## 5. Conclusions

We introduced a brand new survival analysis model and performed survival prognosis on 16 cancer types. The prognosis was significantly improved when self-attention and transfer learning was integrated to the SAVAE-Cox. With the further analysis of the hidden layer features of SAVAE-Cox, we confirmed that the hidden layer features of this model play a significant role in cancer prognosis and the revealing of biological function. In conclusion, the SAVAE-Cox combines a self-attention mechanism with transfer learning, and feature selection provides a new prospect for future deep cancer prognosis.

## Figures and Tables

**Figure 1 cells-11-01421-f001:**
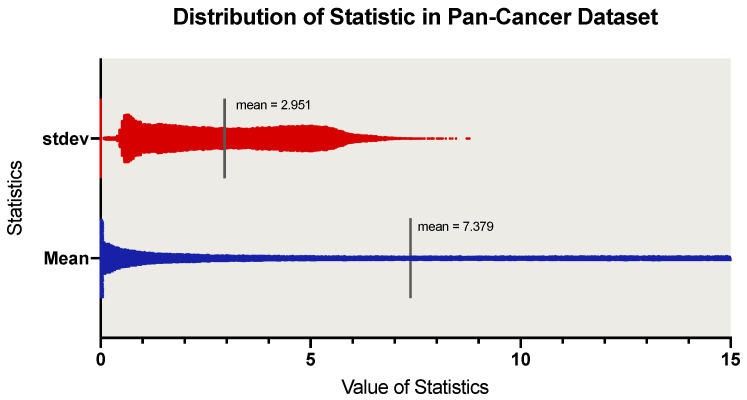
Scatter plot of Pan-Cancer data statistics distribution. The width of the scatter plot represents the number of patient samples. The solid black line represents the mean of the statistic.

**Figure 2 cells-11-01421-f002:**
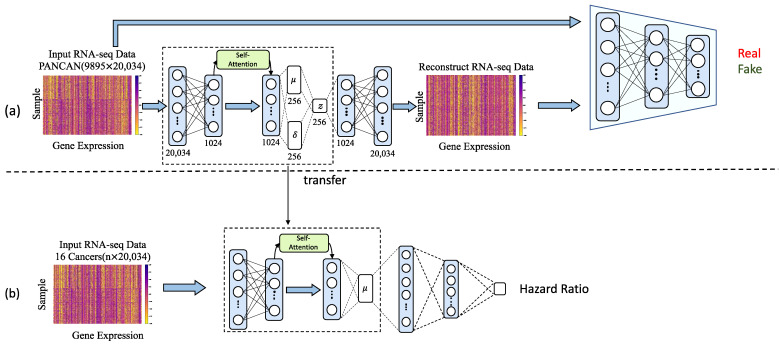
Overview of the SAVAE module. (**a**) Dimensionality reduction pretraining stage using GAN. (**b**) Survival analysis based on transfer learning.

**Figure 3 cells-11-01421-f003:**
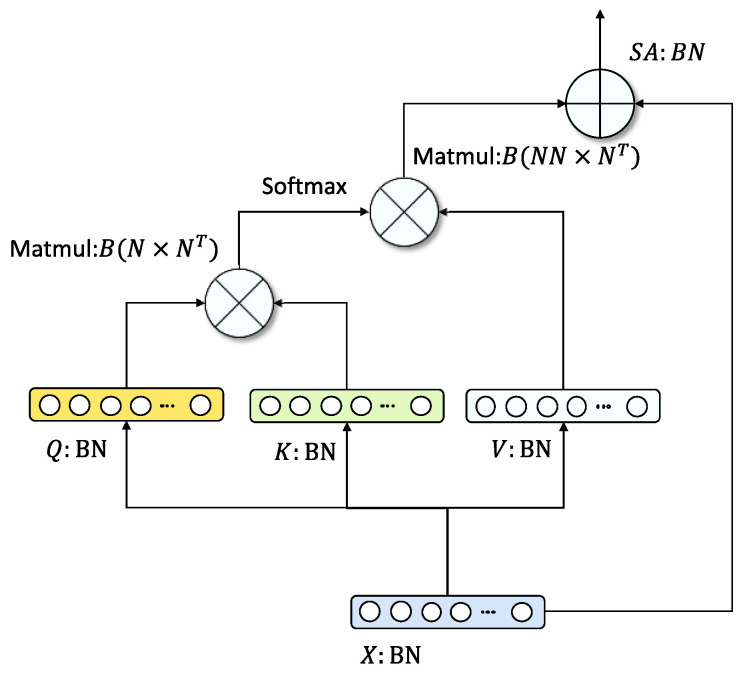
Network framework of residual self-attention module. This network structure can learn the latent semantic correlation of genes.

**Figure 4 cells-11-01421-f004:**
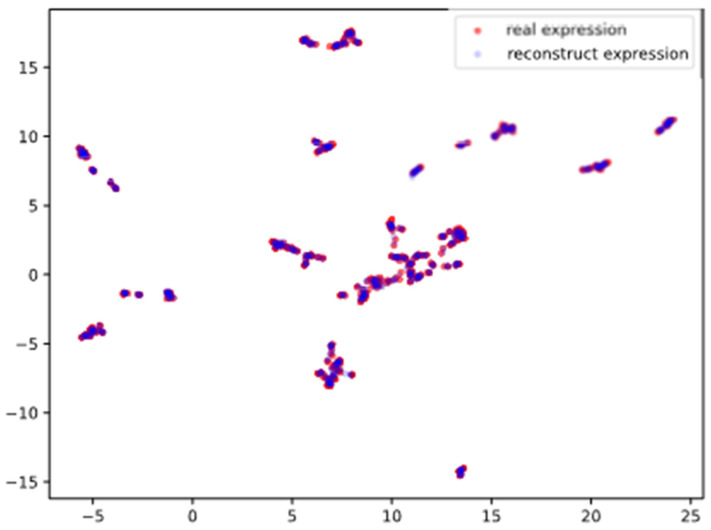
UMap plot of real genes and reconstructed genes. The reconstructed genes and the real genes are highly coincident under low dimension.

**Figure 5 cells-11-01421-f005:**
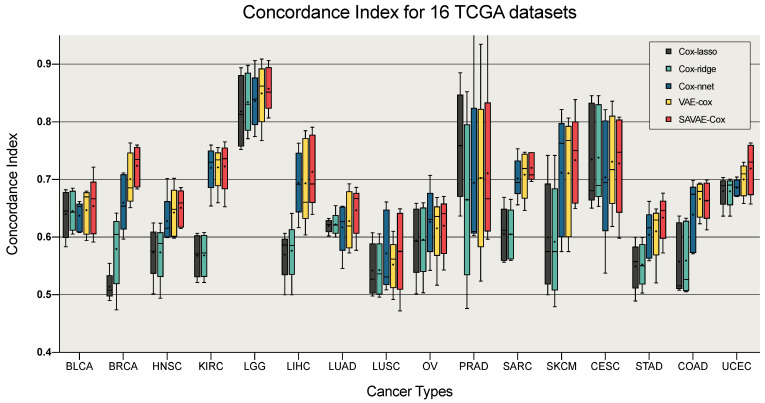
Performance comparison of survival analysis on 16 cancer types. The “+” of each box plot denotes the mean concordance index. The mean concordance index of hazard ratios predicted using our model was best on 12 cancer types.

**Figure 6 cells-11-01421-f006:**
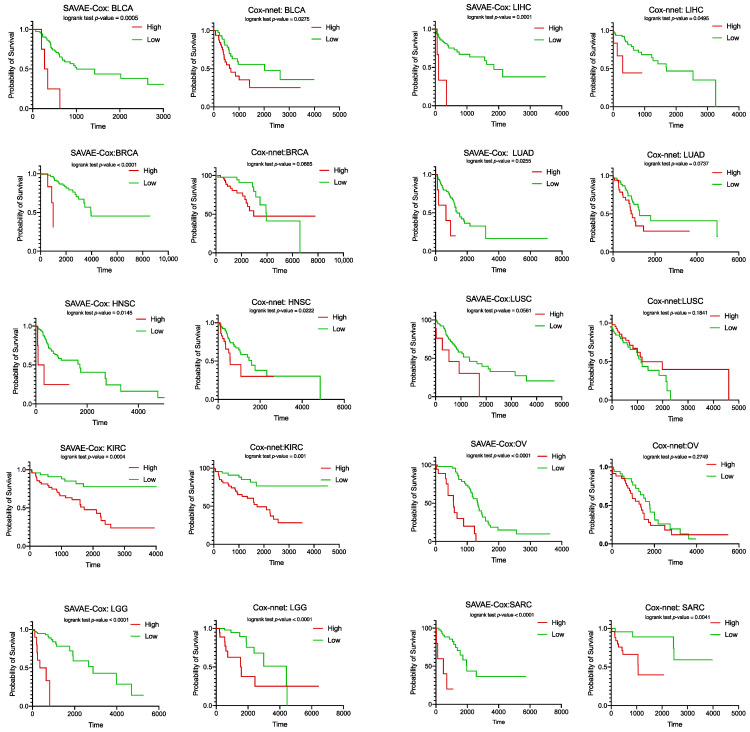
Kaplan–Meier survival curves using SAVE-Cox and Cox-nnet on 12 cancer types. The smaller the *p*-value, the more significant the risk difference between the two groups predicted by the model.

**Figure 7 cells-11-01421-f007:**
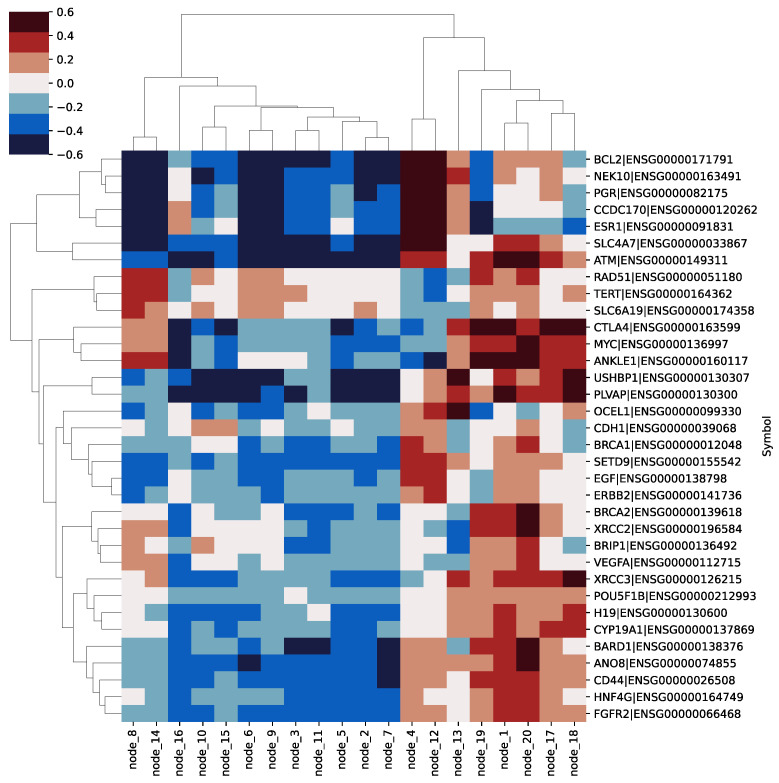
Pearson correlation heatmap of 34 cancer-related genes and 20 key nodes in the BRCA study. All of the 34 genes are highly associated with breast cancer.

**Figure 8 cells-11-01421-f008:**
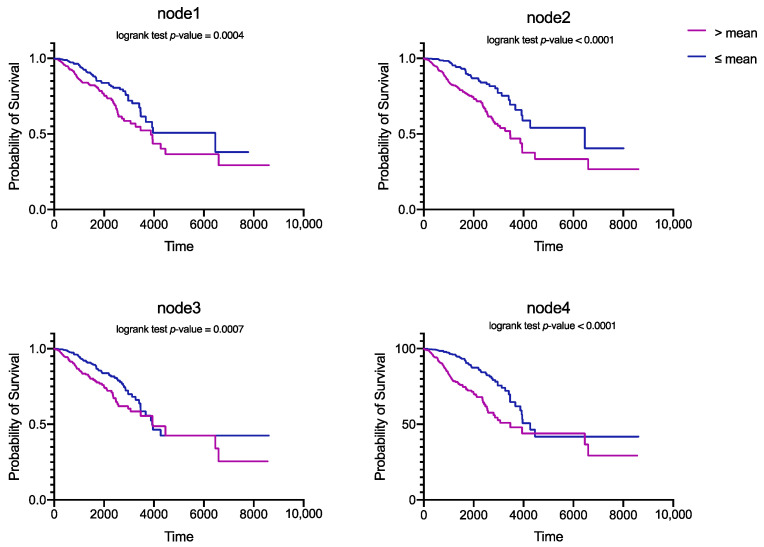
Kaplan–Meier survival curves for four key nodes in a hidden layer. The smaller the *p*-value, the more significant the effect of the node on the survival of the patient.

**Figure 9 cells-11-01421-f009:**
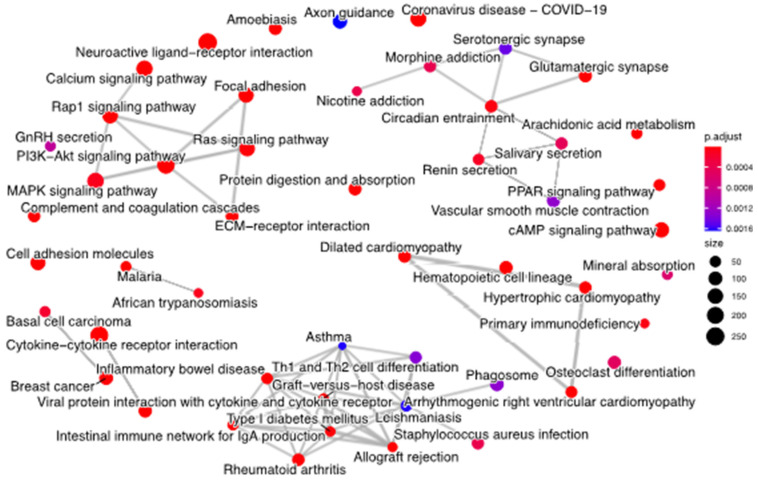
Pathway association network of leader genes. Each point represents a pathway signal, and the gray solid represents the association between pathways. The size of points represents the number of genes enriched in this pathway.

**Figure 10 cells-11-01421-f010:**
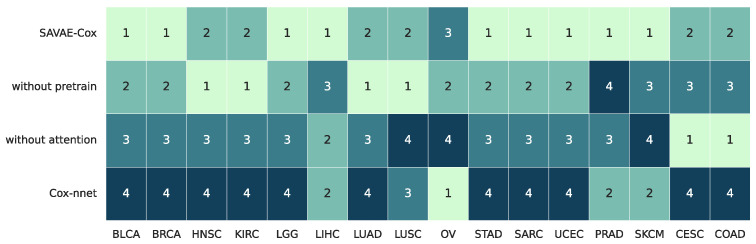
Ablation study on 16 cancer types. The performance results of four models were divided to ranks 1, 2, 3, 4 in descending order.

**Table 1 cells-11-01421-t001:** Statistics of RNA-seq datasets (Pan-Cancer and 16 cancer types) used to train SAVAE-cox.

Cancer Type	Data Attribute		
	Total Samples	Censored Samples	Time Range
PANCAN	9895	#	#
BLCA	397	227	13–5050
BRCA	1031	896	1–8605
HNSC	489	302	1–6417
KIRC	504	347	2–4537
LGG	491	302	1–6423
LIHC	359	183	1–3765
LUAD	491	290	4–7248
LUSC	463	327	1–5287
OV	351	95	8–5481
STAD	345	227	1–3720
CESC	283	215	2–6408
COAD	415	239	1–4270
SARC	253	116	15–5723
UCEC	524	404	1–6859
PRAD	477	289	23–5024
SKCM	312	239	14–1785

#: Not measured in this experiment.

**Table 2 cells-11-01421-t002:** Mean Concordance Index on 16 cancer types using different dimensionality reduction methods.

Cancer Type	Dimensionality Reduction Method						
	AE	Denoise-AE	Chi2	Pearson	MIC	PCA	SAVAE
BLCA	0.642	0.643	0.582	0.552	0.624	0.545	0.654
BRCA	0.704	0.709	0.651	0.500	0.492	0.488	0.724
HNSC	0.649	0.642	0.522	0.590	0.531	0.489	0.651
KIRC	0.725	0.731	0.620	0.698	0.673	0.547	0.723
LGG	0.844	0.843	0.712	0.820	0.786	0.673	0.857
LIHC	0.704	0.696	0.467	0.627	0.401	0.423	0.713
LUAD	0.617	0.635	0.627	0.595	0.571	0.570	0.647
LUSC	0.552	0.559	0.605	0.534	0.529	0.496	0.575
OV	0.608	0.621	0.550	0.512	0.517	0.471	0.620
STAD	0.602	0.616	0.556	0.572	0.531	0.476	0.610
CESC	0.690	0.722	0.724	0.565	0.598	0.398	0.663
COAD	0.631	0.638	0.489	0.533	0.521	0.496	0.728
SARC	0.698	0.700	0.558	0.647	0.637	0.511	0.720
UCEC	0.677	0.701	0.640	0.591	0.611	0.472	0.698
PRAD	0.724	0.649	0.687	0.751	0.586	0.606	0.774
SKCM	0.684	0.655	0.863	0.652	0.531	0.512	0.734

## Data Availability

All datasets used in this study are accessible from UCSC Xena (https://xenabrowser.net/datapages/, Last visited on 21 March 2022). At the same time, we disclosed access links of the 16 cancer types and pan-cancer datasets processed using the method selected in this paper: https://drive.google.com/drive/folders/1KuDVRkPJZWYfQ2Z4YRxDo6e8lo5s68za (Last visited on 21 March 2022).

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
