# Peer review of "A Novel Attention-Mechanism Based Cox Survival Model by Exploiting Pan-Cancer Empirical Genomic Information"

_cells, 2022, doi:10.3390/cells11091421_

Round 1
Reviewer 1 Report
In this study, the authors proposed a novel survival analysis model to improve prognostic accuracy. I have a few comments to improve the quality of the manuscript:
1- The authors need to add more description to the figure captions.
2- The Discussion is very poor and needs to be improved before accepting the manuscript for publication.
3- The authors need to include a section for the conclusions.
Author Response
Response to Reviews’ Comments
Manuscript ID: cells-1669898
Title: “A novel attention-mechanism based Cox survival model by exploiting pan-cancer empirical genomic information”
Dear Editor/Reviewers:
Thank you very much for your great efforts in reviewing our paper! The comments received are very valuable, from which we can feel that the theoretical value of this work is well appreciated, for allowing a resubmission of our manuscript, with an opportunity to address your comments.
Point # 1:
Author response: Thanks for your letter and for the comment concerning our manuscript entitled “The authors need to add more description to the figure captions.” This comment is valuable and very helpful for revising and improving our paper, as well as the guiding significance to our research.
Author action: We accept this comment. We added more detailed descriptions to each figure. Thank again.
Point # 2:
Author response: Thanks for your letter and for the comment concerning our manuscript entitled “The Discussion is very poor and needs to be improved before accepting the manuscript for publication.” This comment is valuable and very helpful for revising and improving our paper, as well as the guiding significance to our research.
Author action: We accept this comment. We discussed the results of SAVAE-Cox in detail in the Discussion section.
Point # 3:
Author response: Thanks for your letter and for the comment concerning our manuscript entitled “The authors need to include a section for the conclusions.” This comment is valuable and very helpful for revising and improving our paper, as well as the guiding significance to our research.
Author action: Thank you very much for your comments, we have included conclusions in the paper and summed up our work in depth.
Sorry for our late response due to the time spent on refining the work, and thanks again for your time and efforts in processing/reviewing our paper!
Best Regards.
Authors
Reviewer 2 Report
Major points
“2.2. Dimensionality reduction pretraining using GAN” Please compare performance of autoencoder and denoising-autoencoder with that of the proposed method.
Instead of dimensionality reduction, feature selection can be used for this study. Please compare performance of the proposed method with those of conventional feature selection methods.
“To validate the pre-trained reconstruction results, we randomly divide the training dataset and the test dataset with a ratio of 8:2.” “All 990 samples in the pan-cancer dataset were tested in this section.” These two sentences can be contradictory.
“Figure 5 represent comparisons of performance on 12 cancer datasets.” Why 12? Table 1 shows 16 cancer types.
“We take BRCA as an example to conduct experiments.” The reason for selecting BRCA is not convincing.
“Data Availability Statement: Not applicable.” I cannot accept this statement.
I hope that the source code of this study is disclosed.
Minor points
“We first downloaded the RNAseq datasets of 16 cancer types in TCGA from the UCSC Xena database” The details of the way to download it should be described as Supplementary material.
“Table 1. Performance comparison on LITS and 3Dircadb datasets” Title of Table 1 is not adequate. Please rename it.
“At the same time, we downloaded the pan-cancer data (PANCAN) RNA-seq data covering 33 cancer types for pre-training of the survival analysis model.” From http://xena.ucsc.edu/public/ ? If so, please clarify it.
“Detailed clinical descriptions of transcriptome data for 16 cancers and pan-cancer data used in this experiment are shown in Table 1.” Pan-cancer data (data of 33 cancer types) is not shown in Table 1.
Please add more meaningful titles in x- and y- axes on Figure 1.
Maybe, Eq (9) is adequate if z is under standard gaussian distribution.
Please clarify the way to determine λ1 and λ2
The detail of training process of SAVAE and SAVAE-Cox is not shown. Epoch, batch size, learning rate, optimizer, etc.
Dimension of z is not shown.
“We believe this model will improve the accuracy of survival analysis.” IMHO, this can be achieved in Figure 10.
Author Response
Response to Reviews’ Comments
Manuscript ID: cells-1669898
Title: “A novel attention-mechanism based Cox survival model by exploiting pan-cancer empirical genomic information”
Dear Editor/Reviewers:
Thank you very much for your great efforts in reviewing our paper! The comments received are very valuable, from which we can feel that the theoretical value of this work is well appreciated, for allowing a resubmission of our manuscript, with an opportunity to address your comments.
Major Point # 1:
Author response: Thanks for your letter and for the comments concerning our manuscript entitled “Please compare performance of autoencoder and denoising-autoencoder with that of the proposed method.” This comment is valuable and very helpful for revising and improving our paper, as well as the guiding significance to our research.
Author action: We accept this comment. Therefore, to better demonstrate the dimensionality performance of our proposed GAN-based dimensionality reduction method, we compare the performance of autoencoder and denoising autoencoder with our method. We provide a detailed analysis of the performance results in Section 3.1 of the manuscript. Meanwhile, we attached the comparison results in Table 2 of the manuscript.
Major Point # 2:
Author response: Thanks for your letter and for the comment concerning our manuscript entitled “Instead of dimensionality reduction, feature selection can be used for this study. Please compare performance of the proposed method with those of conventional feature selection methods.”
Author action: We accept this comment. With reference to your significant concern, we used four classical feature selection methods to compare with our proposed dimensionality reduction method. We attach the comparison results to Table 2 in our manuscript.
Major Point # 3:
Author response: Thanks for your letter and for the comment concerning our manuscript entitled “‘To validate the pre-trained reconstruction results, we randomly divide the training dataset and the test dataset with a ratio of 8:2.’ ‘All 990 samples in the pan-cancer dataset were tested in this section.’ These two sentences can be contradictory.”
Author action: Thank you very much for your concern, there are mistakes in these two descriptions. Because the pan-cancer data has a huge number of samples, we divide the pan-cancer data according to 9:1. The total sample size of pan-cancer data applied to this work was 9895. After the above division method, the training set contains 8905 samples, and the test set contains 990 samples. We revised these descriptions in the manuscript.
Major Point # 4:
Author response: Thanks for your letter and for the comment concerning our manuscript entitled “Figure 5 represent comparisons of performance on 12 cancer datasets. Why 12? Table 1 shows 16 cancer types.”
Author action: Thanks for your comments, we did not update Figure 5 when submitting the manuscript. We re-uploaded the new Figure 5 and changed the description in the text.
Major Point # 5:
Author response: Thanks for your letter and for the comment concerning our manuscript entitled “We take BRCA as an example to conduct experiments. The reason for selecting BRCA is not convincing.” This comment is useful and very helpful for improving our paper.
Author action: We further elaborate on our reasons for choosing breast cancer in our paper. We chose breast cancer as an example for two reasons. First of all, breast cancer is the most common and frequent malignant tumor in the world, and a lot of research are devoted to breast cancer-related research. Up to now, a considerable number of studies have found a large number of breast cancer oncogenes. The use of breast cancer as a case study is well documented. At the same time, benefiting from the professionalism of the TCGA project, the researchers collected abundant samples.
Major Point # 6:
Author response: Thanks for your letter and for the comment concerning our manuscript entitled “Data Availability Statement: Not applicable. I cannot accept this statement.” This comment is useful and very helpful for improving our paper.
Author action: Thank you very much for your comments. We added the link to access the raw data, and we published the dataset preprocessed using the method in our paper.
Major Point # 7:
Author response: Thanks for your letter and for the comment concerning our manuscript entitled “I hope that the source code of this study is disclosed.” This comment is useful and very helpful for improving our paper.
Author action: Thank you very much for your comment. Our source code is available at https://github.com/menggerSherry/SAVAE-Cox. At the same time, we attach the access link to the abstract of the paper.
Minor Point # 1:
Author response: Thanks for your letter and for the comment concerning our manuscript entitled “The details of the way to download it should be described as Supplementary material.” This comment is valuable and very helpful for improving our paper.
Author action: With reference to your comment, we described the download method of the datasets in Appendix A.1 and removed the corresponding statement in the main text. Thanks.
Minor Point # 2:
Author response: Thanks for your letter and for the comment concerning our manuscript entitled “Title of Table 1 is not adequate. Please rename it.”
Author action: Thank you very much for pointing out the mistakes in our manuscript, we have revised the title of Table 1.
Minor Point # 3:
Author response: Thanks for your letter and for the comment concerning our manuscript entitled “‘At the same time, we downloaded the pan-cancer data (PANCAN) RNA-seq data covering 33 cancer types for pre-training of the survival analysis model.’ From http://xena.ucsc.edu/public/ ? If so, please clarify it.” This comment is useful and very helpful for improving our paper.
Author action: The Pan-Cancer dataset is indeed downloaded from Xena, we refer to your comments and clarify it in Appendix A.1. Thanks.
Minor Point # 4:
Author response: Thanks for your letter and for the comment concerning our manuscript entitled “‘Detailed clinical descriptions of transcriptome data for 16 cancers and pan-cancer data used in this experiment are shown in Table 1.’” Pan-cancer data (data of 33 cancer types) is not shown in Table 1.”
Author action: Thank you very much for your comment, our description of this sentence is not clear enough. The Pan-Cancer dataset is used in the SAVAE pre-training stage, so we did not use its clinical data. The Pan-Cancer dataset is from the Pan-Cancer Atlas project. 33 cancer types from the TCGA database were analyzed in this program. UCSC Xena made a preliminary arrangement for this project and created an integrated Pan-Cancer (PANCAN) data set. We only list the statistics of the integrated dataset in the first row of Table 1.
Minor Point # 5:
Author response: Thanks for your letter and for the comment concerning our manuscript entitled “Please add more meaningful titles in x- and y- axes on Figure 1.” This comment is useful and very helpful for improving our paper.
Author action: We accept this comment. We modified Figure 1 and added more meaningful titles in x- and y- axes.
Minor Point # 6:
Author response: Thanks for your letter and for the comment concerning our manuscript entitled “Maybe, Eq (9) is adequate if z is under standard gaussian distribution.” This comment is useful and very helpful for improving our paper.
Author action: We accept this useful comment and further detail the distribution of z in Methods to make Equation 9 more adequate.
Minor Point # 7:
Author response: Thanks for your letter and for the comment concerning our manuscript entitled “Please clarify the way to determine λ1 and λ2” This comment is useful and very helpful for improving our paper.
Author action: We accept this comment. For all hyperparameters used in this work, we search for optimal hyperparameters using empirical knowledge coupled with Bayesian optimization methods. The detailed decision method and optimal hyperparameters used in this work were put in Appendix A.2.
Minor Point # 8:
Author response: Thanks for your letter and for the comment concerning our manuscript entitled “The detail of training process of SAVAE and SAVAE-Cox is not shown. Epoch, batch size, learning rate, optimizer, etc.”
Author action: We described the training process detail of SAVAE and SAVAE-Cox in detail in Section 2.4. And, we list the best hyperparameter settings in Appendix A.2. Thanks again.
Minor Point # 9:
Author response: Thanks for your letter and for the comment concerning our manuscript entitled “Dimension of z is not shown.” This comment is useful and very helpful for improving our paper.
Author action: We set the dimension of z to 256 in this work, we modified Figure 2 in our paper and introduced the dimension of z.
Minor Point # 10:
Author response: Thanks for your letter and for the comment concerning our manuscript entitled “'We believe this model will improve the accuracy of survival analysis.' IMHO, this can be achieved in Figure 10.” This comment is useful and very helpful for improving our paper.
Author action: We removed the description of this sentence, and we elaborated on the accuracy of our method for survival analysis in Section 3.5. Thanks again for your comment.
Sorry for our late response due to the time spent on refining the work, and thanks again for your time and efforts in processing/reviewing our paper!
Best Regards.
Authors
Reviewer 3 Report
The authors proposed an attention-mechanism based Cox survival model for survival analysis of high-dimensional transcriptome. The authors evaluated their tool on 16 types of TCGA cancer RNA-seq data sets and showed that their is superior to some currently available tools.
Overall, the novelty of this work is good, and it will benefit the reader. Important details on datasets used were missing, making it hard to reproduce the results reported. Therefore that should be addressed.
Comments:
- In Section 2.1, there is no specific instruction described for processing the peak data into sequence data usable by the deep learning model. If the data set is previously published along with other methods, a brief description should be included and if multiple datasets from different processing pipelines are used, it's important to identify the difference in the pipeline and their potential impact on the model.
- The compared methods were not fully described in detail. For instance, the values of specific parameters used by those methods were not given. This makes it hard for one to reproduce the results.
- Some evaluation metrics used in the study depend on specific cut-off values to turn the output probabilities into a binary prediction. No details on what the cut-offs are and how these cut-offs are determined were provided.
- As a reviewer I felt that some important works being done in recent times on genome data needs to be discussed as part of CNN being evolutionized for genomic data. Following references can be taken into account,
- https://www.sciencedirect.com/science/article/pii/S2001037021004566
- https://www.worldscientific.com/doi/abs/10.1142/9789811232701_0003
- https://link.springer.com/article/10.1186/s12859-019-2927-x
- https://ieeexplore.ieee.org/abstract/document/9335580
- https://www.worldscientific.com/doi/abs/10.1142/9789811215636_0032
- https://www.nature.com/articles/s41598-020-76759-y
- In figure 4. the umap for real and reconstructed expression should be slightly transparent so that the reader can analyse the similarity in a better manner.
- Its better to include the ablation study.
Author Response
Response to Reviews’ Comments
Manuscript ID: cells-1669898
Title: “A novel attention-mechanism based Cox survival model by exploiting pan-cancer empirical genomic information”
Dear Editor/Reviewers:
Thank you very much for your great efforts in reviewing our paper! The comments received are very valuable, from which we can feel that the theoretical value of this work is well appreciated, for allowing a resubmission of our manuscript, with an opportunity to address your comments.
Point # 1:
Author response: Thanks for your letter and for the comment concerning our manuscript entitled “In Section 2.1, there is no specific instruction described for processing the peak data into sequence data usable by the deep learning model. If the data set is previously published along with other methods, a brief description should be included and if multiple datasets from different processing pipelines are used, it's important to identify the difference in the pipeline and their potential impact on the model.”
Author action: The data used in this work was used GDC Processing Pipeline. We briefly describe the processing flow in Appendix A.1 and attached a detailed explanation of the Pipeline. Thanks again.
Point # 2:
Author response: Thanks for your letter and for the comment concerning our manuscript entitled “The compared methods were not fully described in detail. For instance, the values of specific parameters used by those methods were not given. This makes it hard for one to reproduce the results.” This comment is valuable and very helpful for revising and improving our paper, as well as the guiding significance to our research.
Author action: We accept your comments. We described the method for selecting hyperparameters and showed the detailed hyperparameters in Table A1.
Point # 3:
Author response: Thanks for your letter and for the comment concerning our manuscript entitled “Some evaluation metrics used in the study depend on specific cut-off values to turn the output probabilities into a binary prediction. No details on what the cut-offs are and how these cut-offs are determined were provided.” This comment is valuable and very helpful for revising and improving our paper, as well as the guiding significance to our research.
Author action: We introduce evaluation metrics and clarify cut-off values in Section 2.5. Thanks again.
Point # 4:
Author response: Thanks for your letter and for the comment concerning our manuscript entitled “As a reviewer I felt that some important works being done in recent times on genome data needs to be discussed as part of CNN being evolutionized for genomic data.” This comment is valuable and very helpful for revising and improving our paper, as well as the guiding significance to our research.
Author action: We accept this comment and discussed some important works about CNN based feature extraction module for genomic data in the Introduction.
Point # 5:
Author response: Thanks for your letter and for the comment concerning our manuscript entitled “In figure 4. the umap for real and reconstructed expression should be slightly transparent so that the reader can analyse the similarity in a better manner. ”
Author action: We accept this comment. We modified Figure 4. In the new figure, we adjusted the transparency of the real and reconstructed results to better analyze them.
Point # 6:
Author response: Thanks for your letter and for the comment concerning our manuscript entitled “Its better to include the ablation study. ”
Author action: We did an ablation study for SAVAE-Cox in Section 3.5. We selected four models, Cox-nnet, SAVAE-Cox without pretrain, SAVAE-Cox without attention, and SAVAE-Cox to compare the performance of 16 cancer types and draw the Figure 10. According to the results in Figure 10, we prove that each module of SAVAE-Cox plays a decisive role in the performance.
Sorry for our late response due to the time spent on refining the work, and thanks again for your time and efforts in processing/reviewing our paper!
Best Regards.
Authors
Round 2
Reviewer 2 Report
My concerns have been addressed by the authors in the revision.